# EOG-Based Human–Computer Interface: 2000–2020 Review

**DOI:** 10.3390/s22134914

**Published:** 2022-06-29

**Authors:** Chama Belkhiria, Atlal Boudir, Christophe Hurter, Vsevolod Peysakhovich

**Affiliations:** 1ISAE-SUPAERO, Université de Toulouse, 31400 Toulouse, France; belkhiria.chama@gmail.com; 2ENAC, Université de Toulouse, 31400 Toulouse, France; atlal.boudir@gmail.com (A.B.); christophe.hurter@enac.fr (C.H.)

**Keywords:** human–computer interaction, electrooculography, EOG, human–computer interface, BCI

## Abstract

Electro-oculography (EOG)-based brain–computer interface (BCI) is a relevant technology influencing physical medicine, daily life, gaming and even the aeronautics field. EOG-based BCI systems record activity related to users’ intention, perception and motor decisions. It converts the bio-physiological signals into commands for external hardware, and it executes the operation expected by the user through the output device. EOG signal is used for identifying and classifying eye movements through active or passive interaction. Both types of interaction have the potential for controlling the output device by performing the user’s communication with the environment. In the aeronautical field, investigations of EOG-BCI systems are being explored as a relevant tool to replace the manual command and as a communicative tool dedicated to accelerating the user’s intention. This paper reviews the last two decades of EOG-based BCI studies and provides a structured design space with a large set of representative papers. Our purpose is to introduce the existing BCI systems based on EOG signals and to inspire the design of new ones. First, we highlight the basic components of EOG-based BCI studies, including EOG signal acquisition, EOG device particularity, extracted features, translation algorithms, and interaction commands. Second, we provide an overview of EOG-based BCI applications in the real and virtual environment along with the aeronautical application. We conclude with a discussion of the actual limits of EOG devices regarding existing systems. Finally, we provide suggestions to gain insight for future design inquiries.

## 1. Introduction

Human-computer dialogue has hinged upon input from manual devices (keyboard, mouse, tactile display, etc.) and relies on the visual display—until today. Despite being “ready to deliver the promises” to enhance human–computer interaction for about 20 years now [1], eye-tracking technology use remains narrowed down to applications for disabled users, as eye movements are sometimes their only way to communicate. The gaming industry recently started to adopt eye-tracking to intensify gamers’ experience and enjoyment. Laptops and monitors with integrated eye-tracking devices are available on the market. However, most of these devices are based on video processing, illumination with infra-red light and complex computer vision algorithms. This power-greedy approach is unsuited for prolonged use (energy consumption, produced heat, etc.). Moreover, video-based eye-tracking requires either an integration of a remote camera into an existing environment, which may be difficult, especially in the case of retrofitting; or eyewear, which is mostly incompatible with eyeglasses’ users and party blocks the field of view for the others.

A power-efficient alternative to eye tracking is the application of electrodes on the scalp and face surface known as electro-oculography (EOG). The use of the EOG system was one of the first techniques to analyze eye movements [2], and it has been thought to be a valuable source for communications between humans and computers. One advantage of EOG is that the electrodes avoid the obstruction of the vision. Electro-encephalography (EEG) and electro-oculography are well-suited for brain–computer interfaces and are used to deduce users’ eye movements for passive and active interaction [3]. A complete methodology of EOG based on human–computer interface research is demonstrated in Figure 1. Recent advances in dry [4] and even wearable and flexible textile electrodes [5] increase the comfort of use and the application ease. The recent progress in biopotential signal monitoring lets us imagine embedded EEG and EOG sensors in everyday objects such as a knit cap or an earring. A headset of an airline pilot decoding the fatigue state to adapt the automation level? Or a pair of in-ear headphones changing the music track with a blink of an eye? We propose a literature review of the last two decades’ research on the EOG-based human–computer interaction.

This paper presents a non-exhaustive review of studies published between the years 2000 and 2020 that are based on the EOG eye movement’s interaction in human models. This review provides a precious time- and resource-saving guidebook to exploring the EOG interaction tendency for the last two decades, making it relevant of interest for EOG novices as well for specialists. We included 112 English peer-reviewed journal articles and conference papers that met the inclusion criteria and were published in PubMed, the Institute of Electrical and Electronics Engineers (IEEE), or the Association for Computing Machinery (ACM) digital library. Figure 2 represents the PRISMA flow diagram of the review process. The research of three cited databases was performed in parallel. After the exclusion of non-relevant records, the results were put together and duplicate records removed. The records that were identified contained, in the title and/or abstract the following keywords: (EOG OR electrooculography) AND (interaction OR interface).

The particularity of the used devices, the extracted features, the category of the algorithms and the nature of the interaction (passive or active) of each study have been emphasized in the Table 1 and Table 2. Figure 3 indicates the number of included publications per 5-year periods. It shows the increasing interest in the technology. Figure 4 illustrates the distribution of different studies characteristics per 5-year period.

## 2. Acquiring EOG Signal

The eye is an important source of the relevant information associated with the user’s activities and cognitive functions. It can be modeled as an electrical dipole (between the retina and the cornea) that follows eye movements. If we place two electrodes near two eyes’ corners and two electrodes on the top and bottom of one eye, there will be a voltage difference between the vertical and horizontal electrodes, which is known as the EOG signal. When the eyeball moves in the direction of the electrode, the electrical potential increases, and if the eye moves in the other direction, the electrical potential decreases. The EOG signal is a measure of the corneal–retinal potential difference with signal amplitude ranging 15–200 μV. Using an adequate calibration procedure, the signal can be mapped to the eyeball angular position with a 2-degree range in the vertical direction and a 1.5-degree range in the horizontal one [6,7,8].

Interestingly, EOG signals can be obtained using cheap and simple electrodes. Generally, horizontal electrodes are placed on the outer boundary of the eye, while vertical electrodes are placed above and below the eye. The reference electrode is often positioned on the forehead [9]. Other alternative electrode placements have been broadly described in our former review [3]. The EOG signal is proportional to the eye gaze move and it is commonly distinct from other bioelectric potentials. Concerning hardware requirements, EOG signal acquisition is based on cheap equipment with extensive eye-tracking capabilities allowing the field of view to not be limited to cameras or sensors.

## 3. EOG Devices

There are different EOG devices found across our literature review and which are summarized in the Table 1 and Table 2. Four categories were distinguished: standard EOG, alternative EOG mounting, J!NS MEME glasses and customized EOG devices.

(i)The standard EOG consisted of the placement of two electrodes at the left and right corners of the participant’s eyes to measure the horizontal movement. Two other electrodes were placed up and down of one eye to obtain the vertical movement. A ground electrode was positioned on AFz (as required by the International 10/10 System) and the reference electrode was positioned on the right or left earlobe [10].(ii)The alternative EOG mounting represented an uncommon position of surface electrodes such as around the ears. Manabe et al. [11] placed four electrodes on the head at the locations of the common headphone cushion. Ang et al. [12] used an alternative EOG system where activities were recorded by a single-channel commercial headset NeuroSky MindWave Mobile Headset (NeuroSky, San Jose, CA, USA). The device was formed by a single-channel sensor based on a dry electrode with stainless steel. The sensor was placed on the participant’s forehead to capture electrical signals produced by the brain and the muscles.(iii)The J!NS MEME glasses (JINS MEME ES Digital Innovation Lab, Fujitsu, Tokyo, Japan) employs 3-point stainless steel EOG electrodes on the nose bridge and IMU, battery, and Bluetooth units on the eyeglasses temples. The J!NS MEME are not a common computing interface but rather a sensing device. They stream the data from the sensor to smartphones or laptops via Bluetooth. The sensor data corresponds to vertical and horizontal EOG signal, accelerometer and gyroscope data. The equipment has an operating time of 8 hours. It allows a long duration of real-time streaming of eye and head movements. The J!NS MEME are unobtrusive and look like ordinary glasses.(iv)The customized EOG device category presents the mounted system elaborated by the experimental group according to their objectives. Here we introduce a brief presentation of each customized device included in this review. Vehkaoja et al. [13] used a prototype of a wearable and wireless device designed for EOG and also for facial electromyography (EMG) data recorded from the participant’s forehead. The device was based on five easy-to-proceed textile electrodes that were inserted into a head cap. Bulling et al. [14] employed a wearable and standalone device. It was formed by dry electrodes integrated into goggles with a small pocket-worn constituent with a digital signal processor to explore EOG signals in real-time. Bulling et al. [15] recorded EOG data from the commercial Mobile Brain/Body Imaging (MoBI) from Twente Medical Systems. The device was formed by four-channel EOG that was worn on a belt around the participant’s waist and data were transmitted via Bluetooth. Kuo et al. [16] captured the horizontal eye-gaze direction to control wheelchair driving. They used a pair of eyeglasses to set up left and right surface electrodes as a compact modular. Zhang et al. [17] developed a wireless and lightweight head-mounted system that measured both EOG and EEG for interacting in a virtual reality environment. They positioned six horizontal and vertical electrodes symmetrically around both eyes. This device captures azimuth, elevation and vergence of gaze. Xiao et al. [18] proposed a single-channel EOG device that enables real-time interactions with the virtual reality environment. They designed a graphical user interface for the EOG-based BCI in virtual reality that contains several buttons. The user had to blink while the algorithm identifies the eye blinks and detects the user’s target button. Vidal et al. [19] measured eye movements’ amplitude and duration in reaction to the stimulus with an elaborated experimental system consisting of three devices. First, they connected EOG electrodes to the Mobile Brain/Body Imaging (MoBI) approach developed by Scott Makeig’s group. Second, they implemented an infrared eye tracker from Ergoneers GmbH. Third, they recorded head movements by connecting a cap with an inertial quantification. Inaez et al. [20] integrated five EOG electrodes into a pair of glasses. They incorporated electronic and mechanical elements including a printed circuit board, batteries, communication module electrode holders, lenses and frame. English et al. [21] analyzed four electrodes from the EEG Emotive EEG Headset (San Francisco, CA, USA) to process small and large amplitude of EOG signals. Their conceptualized EOG eyephone system was efficient during sitting, standing and walking. Valeriani et al. [22] recorded EOG signals utilizing two facial electrodes placed on the forehead. The interaction was realized through eye winks by comparing the peak amplitudes and an app installed on the smartphone. Kosmyna et al. [23] included EEG and EOG electrodes, an amplifier, a Bluetooth module and a speaker for bone conduction auditory feedback in a pair of glasses. The operator rather received feedback or nudges sent by a wireless vibration brooch. Despite the varied and diverse acquisition devices, the different investigations shared the next step which concerns features extraction.

Figure 4A shows that the most used electrode placement is the standard scheme. It also shows that there is a recent increasing interest in using J!NS MEME glasses. It might show that it is a off-the-shelf product that allows the use of EOG technology without any wire nor sticking the electrodes.

## 4. Extracted Features

EOG signal is commonly used as a human–computer interface tool to control signals [24] and plays a crucial role in the description and classification of eye movements. EOG is used in a diversity of applications, such as eye writing ordering [25], electric wheelchair control [26], cursor mouse selection [27], eye movement recognition [28] and mobile robot direction [29]. To achieve high performance in identifying and classifying EOG signal, several methods have been presented, including derivative technique [30], threshold evaluation [31], slope analysis method [32] and peak detection exploration [33].

Feature extraction methods and analysis of EOG artifacts are one the most important challenges in this field. The study of Aungsakul et al. [34] evaluated 14 EOG features based on a class separation point of view. All the extracted features were performed on the time domain toward the simplification of the calculations. They used both horizontal and vertical EOG channels and they combined seven methods: maximum peak amplitude value, maximum valley amplitude value, maximum peak amplitude position value, maximum valley amplitude position value, areas under a curve, number of threshold crossing values, and variance of EOG signal. All of these extracted features showed a statistical significance with *p* < 0.0001. To extract EOG features, Pournamdar et al. [35] explored the extreme point strategy that considers the time of occurrence of maximum and minimum values. The polynomial fitting method and eye movement artefacts have been extracted and categorized.

Vidal et al. [19] developed a feature-based method that allows differentiating between saccades, smooth pursuits and vestibulo–ocular reflex movements. They extracted features related to range, velocity, and acceleration from raw and filtered EOG signals: mean velocity, maximum velocity, mean acceleration, maximum acceleration, range of amplitude, mean velocity, maximum velocity, mean acceleration, maximum acceleration, range of amplitude, coefficients of polynome fitted to the signal and the slope of the signal. Even though they did not include fixation and blinks in their model, they were among the first groups to analyze eye movement features to provide a base of eye movement recognition algorithms for exploration in mobile environments.

The number and quality of the signal extracted features are affected by the degree of freedom of the end effector. For instance, controlling a computer mouse or robotic arm needs 2 to 4 degrees of freedom [36], while achieving the dexterity of a single joint of the hand may require up to 22 degrees of freedom. The advantage of the EOG system is that is well suited for applications with limited degrees of freedom, although it is difficult to extract a large number of control signals from the average neuron population activity and may limit the scalability of such a technique.

Feature extraction allows obtaining an optimal subset of elements to classify a selection of collected data according to the criterion functions. Nowadays, an important number of robust and flexible classification algorithms have been developed to recognize and classify the extracted features from EOG signals.

Figure 4B shows that most studies are using blinks and saccades as features. No particular trend can seen.

## 5. Classification Algorithms

To classify the EOG signal extracted from the eye movements, several algorithms are often used to identify the eye movements classification including saccades, blinks, fixations, smooth pursuits and artifacts removal. EOG signal data are predominantly detected in low-frequency bands. Generally, the band-pass filter ranging from 0.1 to 30 Hz with a sampling rate of 128 Hz is performed. After that, a moving average filter is used to eliminate the existing noise components. The eye movements classification is based on an algorithm that computes the derivative of the EOG signal. The EOG derivative is measured through the amplitude value of the EOG signal. Two values are needed, one representing the previous value of the amplitude (*n* samples before), e(t−n), and the other representing the current value, e(t) [37].

An important number of algorithms were designed and employed in several studies to classify eye movement signals. Each decoding algorithm is unique to each BCI and needs to be adaptable to new users, and can self-calibrate. EOG mutual adaptation of “user-to-system and system-to-user” is unique to BCI. The k-nearest neighbors’ algorithm is particularly selected in brain–computer interface investigation for its relevant outcomes. It is one of the simplest and most efficient algorithms for the classification of samples. The procedure requires firstly detecting the k parameter (corresponding to the number of neighbors). After that, the distance of the unknown sample is designed with all known samples. When the samples are categorized according to their distance and value of k, the class that has the majority of nearest samples is chosen as the class of unknown samples (L. Pan 2015). One of the several versions of this algorithm is called random sub-sampling [38].

The Time Delay Neural Network is another dynamic network that classifies EOG signals. The input delay feedforward feedback neural network is a time-delay neural network, and its hidden and output neurons will be copied over the entire time. The delay is computed from top to bottom, so the network has a tapped delay line, which can connect the current signal, the previous signal and the delayed signal to the network weight matrix through the delay time unit (such as 0, 1 and 2). To identify the eye patterns, the Time Delay Neural Network replicated the former activation at each step and updated the outgoing connection with the original unit. These units are coupled to the next layer called the receiving field. The Time Delay Neural Network uses the Levenberg backpropagation training algorithm where the samples are normalized from 0 to 1 with a binary normalization algorithm [39].

Other algorithms contain filtering, feature extraction, training and actual event detection stages to detect eye movements. The minimum redundancy and maximum relevance algorithm allows the number of features to be reduced by selecting the most relevant ones. Clearness Based Feature Selection calculates the distance between the target sample and the center of each category and then compares the class of the nearest center with the class of the objective sample [40]. The eye movements have non-repetitive and unstable features. The EOG signal processing is often challenging to denoise and classify. Therefore, an algorithm-based classification that requires both structural and temporal analysis is constantly being developed.

Figure 4C shows that most studies are using linear algorithms. It also shows that the majority of recent studies indicate the used algorithm that was not the case in the years 2000–2005.

**Table 1 sensors-22-04914-t001:** Years 2000–2015. Pair rows are highlighted in gray for the ease of the reading.

													Interaction					
Reference		Device				Extracted Features				Algorithm			Active				Passive	
**First** **Author**	**Year**	**Custom Device**	**J!ns MEME**	**EOG Standard**	**EOG Alternative**	**Blinks**	**Saccades**	**Eye-related**	**Other Features**	**Linear**	**Network**	**NM**	**“Keyboard” Input**	**Navigation**	**Conf./Select/Point**	**Visually Guided** **?**	**Fatigue/Cognitive**	**Activity**
Chen [41]	2000	•			•					•						•		
Rosander [42]	2000		•		•					•						•		
Barea [43]	2002	•				•				•		•						
Ding [44]	2005	•				•								•				
Vehkaoja [13]	2005	•					•					•						•
Manabe [11]	2006		•	•	•	•		•					•					
Ding [45]	2006	•			•	•		•					•					
Trejo [46]	2006	•				•			•				•					
Yamagishi [31]	2006	•		•					•		•							
Akan [47]	2007	•		•	•								•					
Bashashati [48]	2007	•		•				•								•		
Krueger [49]	2007	•				•		•					•					
Skotte [50]	2007	•		•				•						•				
Estrany [51]	2008	•			•					•			•					
Bulling [14]	2008	•				•	•				•			•				
Cheng [52]	2008	•			•			•						•				
Mühlberger [53]	2008	•					•			•					•			
Bulling [15]	2009	•				•	•	•		•					•			
Estrany [54]	2009	•		•					•					•				
Bulling [55]	2009	•		•	•				•							•		
Kuo [16]	2009	•					•			•				•				
Zheng [56]	2009	•		•	•			•					•					
Keegan [57]	2009	•			•				•				•					
Usakli [58]	2009	•		•	•					•	•							
Yagi [59]	2010	•		•				•					•					
Usakli [60]	2010	•			•	•					•							
Belov [61]	2010	•			•			•								•		
Zhang [17]	2010	•						•								•		
Punsawad [62]	2010	•				•		•						•				
Vidal [19]	2011	•				•	•	•		•								•
Bulling [28]	2011	•		•	•				•					•				
Li [63]	2011	•					•								•			
Liu [64]	2011	•		•				•			•							
Banarjee [37]	2012		•		•	•		•				•						
Tangsuksant [65]	2012	•		•	•	•		•			•							
Swami Nathan [66]	2012	•		•	•			•			•							
Iáñez [20]	2013	•		•		•	•			•					•			
English [21]	2013	•				•	•	•	•	•					•			
Ubeda [67]	2013	•			•	•		•					•	•				
Li [68]	2014	•		•		•				•					•	•		
Manabe [69]	2014	•						•		•						•		
Ishimaru [70]	2014		•			•				•								•
Witkowski [71]	2014	•					•			•					•			
Ma [72]	2014	•				•		•	•	•				•				
Jiang [73]	2014	•			•	•		•					•					
Wang [74]	2014	•		•								•						
Aziz [75]	2014	•				•		•				•						
Hossain [76]	2014	•		•	•			•					•					
D’Souza [77]	2014	•				•		•	•							•		
Manmadhan [78]	2014	•		•	•			•					•					
OuYang [79]	2015	•		•	•	•		•								•		
Manabe [80]	2015	•				•		•								•		
Ishimaru [81]	2015		•			•	•			•					•		•	•
Hossain [82]	2015	•		•	•			•			•							
Valriani [22]	2015	•			•	•				•					•			
Banik [83]	2015		•		•								•					
Ang [12]	2015		•	•				•					•					

**Table 2 sensors-22-04914-t002:** Years 2016–2020. Pair rows are highlighted in gray for the ease of the reading.

													Interaction					
Reference		Device				Extracted Features				Algorithm			Active				Passive	
**First** **Author**	**Year**	**Custom Device**	**J!ns MEME**	**EOG Standard**	**EOG Alternative**	**Blinks**	**Saccades**	**Eye-related**	**Other Features**	**Linear**	**Network**	**NM**	**“Keyboard” Input**	**Navigation**	**Conf./Select/Point**	**Visually Guided?**	**Fatigue/Cognitive**	**Activity**
Kumar [84]	2016	•				•				•					•			
Dhuliawala [85]	2016		•					•		•					•			
Shimizu [86]	2016		•			•	•			•						•		
Shimizu bis [87]	2016		•					•		•				•	•			
Bissoli [88]	2016	•				•		•					•					
Wilaiprasitporn [89]	2016	•		•					•						•			
Fang [90]	2016	•			•	•		•			•							
Tamura [91]	2016	•		•		•	•			•			•		•			
Barbara [92]	2016		•			•	•			•			•					
Atique [93]	2016	•		•	•						•		•					
Naijian [94]	2016	•						•							•			
Ogai [95]	2017	•			•	•		•					•					
Lee [96]	2017		•						•	•								•
Robert [97]	2017		•				•			•								•
Ishimaru [98]	2017		•				•				•							•
Kise [99]	2017		•				•			•								•
Augereau [100]	2017		•				•			•				•				
Tag [101]	2017		•			•						•		•				
Thakur [102]	2017	•				•			•							•		
Chang [103]	2017	•			•	•					•							
Huang [104]	2017	•		•									•					
López [25]	2017	•		•	•						•							
Heo [105]	2017		•	•	•	•		•			•	•						
He [106]	2017	•		•				•			•							
Lee [107]	2017	•		•	•			•			•							
Zheng [56]	2017		•	•	•			•							•			
Zhang [108]	2017	•		•		•		•					•					
Zhi-Hao [109]	2017	•		•									•					
Hossain [27]	2017	•			•			•					•					
Soundariya [110]	2017	•				•		•							•			
O’Bard [111]	2017	•			•			•			•	•						
Perin [112]	2017	•			•	•		•				•						
Karagöz [113]	2017	•		•	•			•										
Crea [114]	2018	•																
Zhang [115]	2018	•		•									•	•				
Lee [107]	2018	•		•							•			•				
Kim [116]	2018	•			•	•					•							
Fang [117]	2018	•			•	•			•		•							
Bastes [118]	2018	•		•		•	•	•				•						
Hou [119]	2018	•		•	•			•			•							
Jialu [120]	2018	•			•	•			•		•							
Sun [121]	2018	•		•	•	•					•							
Xiao [18]	2019	•				•				•					•	•		
Lu [122]	2019	•		•				•					•					
Garrote [123]	2019	•		•	•				•			•						
Tag [124]	2019		•						•	•							•	
Findling [125]	2019	•					•			•			•					
Rostaminia [126]	2019		•			•					•							•
Kosmyna [23]	2019	•				•	•				•						•	
Badesa [127]	2019	•				•		•					•					
Zhang [128]	2019	•		•	•							•						
Wu [129]	2019	•		•											•			
Mocny-Pachońska [130]	2020		•			•	•	•	•	•							•	
He [131]	2020	•		•				•					•					
Huang [132]	2020			•		•		•					•					

## 6. EOG-Based Interaction

Figure 5 presents the percentage of each type of interaction based on our investigation of the published EOG-based BCI works of the last 20 years. The interactions can be generally classified into active and passive [133]. During active interactions, the EOG signal is intentionally used to control the system by the user. During passive interactions, EOG signal is used to monitor the user’s state without explicitly controlling the system. According to our observation, active interaction contributed with 76% while passive interaction contributed with only 24% of the total included number of EOG-based BCI applications. The active interaction was distributed between 43% for pointer/selection, 25% for visual keyboard, 17% for web navigation and 15% for monitor visual guide. The passive interaction was divided between 62% of activity recognition and 38% of cognitive demand. The important involvement of the active interaction can be considered as an important indicator of EOG signal simplicity. In contrast, passive interaction may require more complicated feature extraction and classification approaches.

The central role of BCI systems is to transduce the neural signal from the brain into signals understandable by an external piece of hardware. The interface implements direct communication access between the brain and the device to be controlled. The EOG system has the potential to create active or passive user interaction with the environment. The interaction based on EOG is explored in an important number of applications, including eye writing commands [25], eye movement recognition [24], wheelchair drive [26], pointer mouse selection [27] and movable robot activity [29].

The nature of the user interaction depends on the target and the field of application including rehabilitation (e.g., exoskeleton), virtual reality, response to physical stimuli (e.g., pain), or identification of brain states (e.g., fatigue or frustration). The EOG similar to other BCI systems, allows the user to interact and control different environments and actions, such as computer function (e.g., word writing and web browsing), mobility conduction (e.g., wheelchair driving) and daily environmental manipulation (e.g., light, television). Interestingly, EOG can be explored as a BCI tool for the therapeutic purpose to rehabilitate normal visual sensory system control by generating activity-dependent neural plasticity.

Figure 4D shows the distribution of the used interactions per 5-year period. It shows that in the years 2000–2005, the dominant interaction was passive detection of user’s activity. Recent studies suggest a trend towards using EOG as keyboard-like input, and increasing interest in using EOG as passive method to determine the level of fatigue. Eventually, the most popular interaction is for actions of confirmation, selection, and pointing.

### 6.1. Real Environment Interaction

Vidal et al. [134] suggested considering smooth pursuit data to analyze calibration-free gaze interaction. Their method is not based on the gaze position, as the traditional gaze interaction methods are, but it associates smooth pursuit with targets that move dynamically on the interface. A hands-free and calibration-free interactive technology were presented by Esteves et al. [135] for smartwatches. It is based on smooth pursuit targets that rotate on the interface along a radial trajectory.

Kangas et al. [136] analyzed visual, auditory and haptic feedback methods to achieve smooth pursuit gaze tracking. They found a significant user preference for haptic and audio feedback. Another study was conducted by Khamis et al. [137] on spontaneous gaze-based interaction when users are displaying pursuits. The study of Jalaliniya et al. [138] presented a method to recognize objects of interest only by analyzing the movement of the eyes while presenting visual stimuli that move horizontally or vertically. Schenk et al. [139] exposed a gaze interaction method based on eye movements for static user interfaces.

### 6.2. Virtual Reality Interaction

EOG is being studied as BCI for virtual and augmented reality, particularly in the case of gaming, rehabilitation, or for training and evaluation of motion, image-based decoding [140,141]. Kumar and Sharma [84] introduced an EOG hands-free natural system that increases the involvement in virtual games. Six EOG electrodes were used for signal acquisition and seven directions were realized by eye movements (left, right, up, down, gaze, wink and double-blink). One of the advantages of combining the EOG system with a virtual environment is to enhance the opportunities for users to participate in retraining by increasing visual feedback. Another benefit is the rapid recruitment of cortical motor networks throughout the motion images to control the interaction. Moreover, EOG interaction in virtual reality can help promote the participation of motor learning mechanisms through repeated training, even if the user can control their limbs [142].

In addition, the importance of the EOG-based HCI for interaction in a virtual reality environment concerns the following points: Firstly, the EOG-based HCI hardware system is effortless and wearable. Then, the EOG-based HCI presents gratifying efficacy. Third, several virtual reality applications might be applied through the EOG-based HCI. A prominent method of virtual reality interaction is gesture tracking which includes two types: optical tracking (e.g., Leap Motion or Nimble VR depth sensors) and the sensors positioned on the hand (e.g., data gloves) [143]. Although an important number of works prove the flexibility and attractiveness of eye movements for natural and uncalibrated interaction, they are still video-based eye-tracking.

### 6.3. Aircraft Pilot Interaction

In the aviation field, it is known that errors related to sleepiness and fatigue in the cockpit lead to fatal accidents [144]. Even in the military environment, pilot fatigue can lead to the inability to invest in enemy targets, mistargeting, or to bombing familiar resources in the air or on the ground. Caldwell et al. [144] showed that even professional Air Force fighter pilots are vulnerable to the effects of fatigue during prolonged waking periods. Wilson et al. [145] recorded electrophysiological pupil area data to evaluate fatigue during task performance in aviation. They showed concomitant and decremented performance changes and they emphasized the importance of brain interaction in this environment. Increasing pupil area was associated with increased task difficulty which is consistent with a large body of literature demonstrating the correlation between increasing pupil diameter and cognitive task loads [146]. One obstacle with interpreting the eye data results is the variability of the light levels from the operator interface during the flight. Low and high luminance conditions may impact the cognitive changes in pupil diameter [147].

One advantage of EOG-based interaction is that the light conditions have only little effect on the EOG signal, which is principally important for mobile recording such in daily life or even in extreme environments. Contrary to video-based eye-tracking, EOG assessments do not require light to come into the eyes to trigger the electrical potential. EOG measurement can be realized in obscurity or even with closed eyes. Due to this feature, EOG is particularly used for eye movement measurements during normal [148] and pathological sleep such detection [149] that can be applied to analyze pilots’ sleepiness.

Another potential benefit of EOG for application in aviation domain is the light-weight computational analysis. It is not based on complex video and image processing. Therefore, EOG-based interaction can be commonly used in a laboratory environment [150], as well as a completely mobile system operable during simulated or real flight. EOG-based systems offer also the possibility to perform mobile long-term recording that does not require additional apparatus for data recording and storing allowing the live capture of pilots’ eyes activity during real flight situations [151,152].

## 7. EOG Limits

Although EOG-based BCI systems have made important advances in exploring eye movement signals for interactions, the methods show some complications with an important number of applied operations. A limitation of the conventional EOG-based BCI paradigm is that the operators may be willing to adjust the eye movements for few seconds by fixing the gaze at an accurate position. When compared with video-based tracking, EOG has the limitation to be based on the around-eyes electrodes that are attached to the skin. EOG signal generates a simple waveform, a linear correlation with eye movement and it produces a considerable signal-to-noise ratio. Furthermore, EOG delivers worse spatial point of gaze tracking accuracy so it is more suitable for relative eye movements’ analysis. EOG signals are also susceptible to artifacts and drifting, mainly when recorded with a mobile system. Similar to other physiological signals, EOG signals are often corrupted by noise from the electrical activity of the body, electrodes, residential power lines, measuring circuits and other interference physiological sources. In addition, similar to EEG, EOG suffers from poor spatial resolution such can be obtained from other neuroimaging techniques. Contrary to eye-tracking systems using videos based on infrared cameras, the EOG-based BCI systems do not use complementary systems. However, the EOG requires the operator’s voluntary movement of eyeballs. Therefore, it is complicated for participants with oculomotor impairments to guide their eyeballs during a basic psychological paradigm such as the orientation of their attention in the direction of a sound source. Finally, EOG data, such as general data processing, deserves specific care before being analysed. As an initial step, data cleaning must be applied with a signal detrending. To do so, a bandwidth filtering can be applied. This will remove high frequency noise but also the continuous component of the signal. Since EOG signal have low voltage amplitude, they are very sensitive to external electrical context. For this reason, extra filtering is usually recommenced with the standard electrical frequency wave removal (50 or 60 Hz). Next, artefacts such as blinks, or parts of the signal that cannot be analysed have to be removed. For instance, when the user speaks, the activation of the jaw muscles spoils the signals and generally resulting into altered collected data. Other data processing stages may be required depending of the recording context and we only give here the most general ones.

## 8. Conclusions

The use of the EOG signal is emerging as one of the most successfully explored bioelectric signals in the BCI domain, and this review showed an increasing interest over the last 5 years. We included 112 relevant studies after the screening of PubMed, ACM DL and IEEE Xplore databases. We presented the results as an overview table according to four design space dimensions: used device, features extracted from the signal, algorithm used for features computation and designed signal-based interaction. We also compared the trends in studies’ distribution per 5-year period. The results showed that the most used electrode placement is the standard scheme, but there is a recent increasing interest in using an off-the-shelf product that allows using EOG technology without wires or sticky electrodes setup. The majority of studies use blinks and saccades as features without any particular trend over the last years. The most popular detection algorithms are the linear ones due to their ease of use for real-time applications. The results of the designed interaction show that while early studies investigated the general passive activity recognition application, recent studies tended to design EOG-based keyboard-like interactions. Eventually, the most popular interaction is for actions of confirmation, selection and pointing.

We also presented the EOG based on BCI applications in the real and virtual environments along with the aeronautical application. Finally, we exposed the limits of the EOG devices regarding existing systems and user perspectives regarding BCI technology. We conclude that almost all the designed EOG systems are highly promising in terms of performance and do not require heavy signal processing and complex pattern recognition algorithms such as those used with other systems as EEG. Nevertheless, most of the EOG-based BCI systems are still based more on theoretical rather than practical models and are suffering from the lack of concrete practice, particularly mobile EOG. Mobile EOG-based BCI systems are considered a relatively new research topic. An important number of investigations should be carried out to explore innovative protocols with mobile interaction control. An efficient system should constantly strive to improve speed and accuracy to eliminate the operator’s frustration and to enable a stable performance level. In addition, the new-generation models may continually take into consideration the user experience, which may reduce the physical and mental stress.

## Figures and Tables

**Figure 1 sensors-22-04914-f001:**
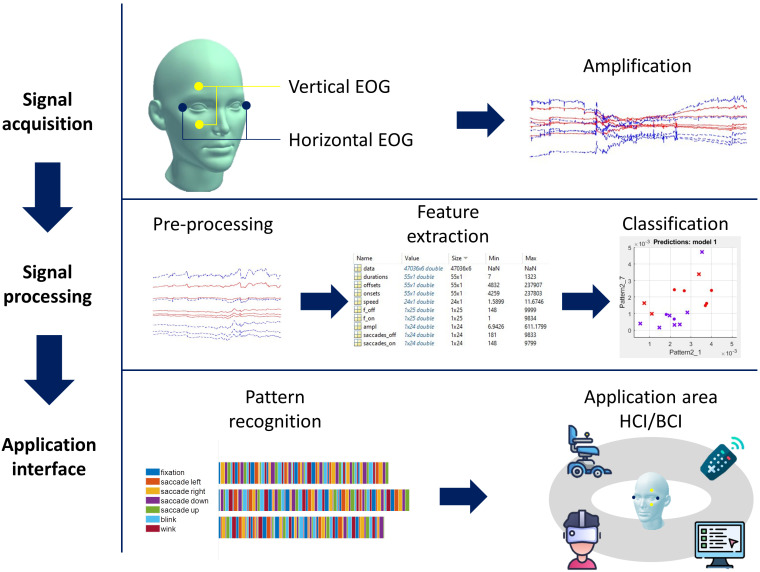
General scheme of an EOG-based BCI. First step, EOG acquisition: EOG is recorded using vertical and horizontal electrodes by an amplified ADC. Second step, Signal processing: after pre-processing, computer processing extracts the most relevant features for identifying the subject’s intentions. Third step, Application area: When the command is recognized and the intention is classified, the instruction is sent to an external device (e.g., web browser, wheelchair, or text display). Feedback informs the user of the results of their actions to allow them to prepare for the next command.

**Figure 2 sensors-22-04914-f002:**
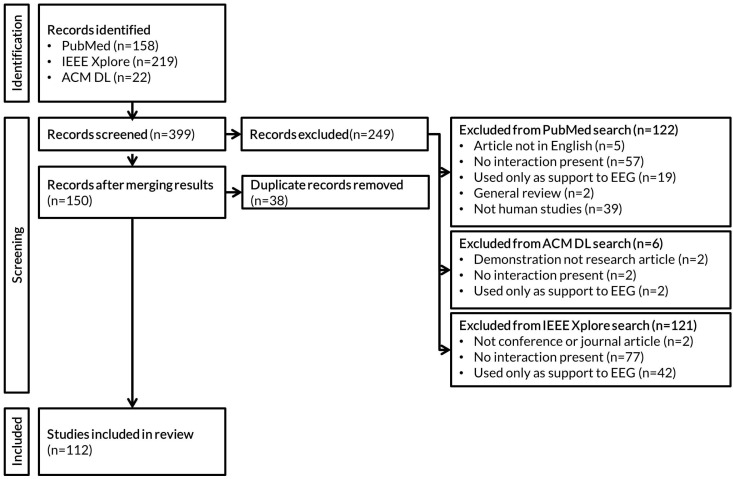
The PRISMA flow diagram of the review process. Please note that, given the work organisation during the project, the review of the databases was paralleled, and, therefore, the removal of duplicate records was done after the non-relevant records were removed from each database list separately.

**Figure 3 sensors-22-04914-f003:**
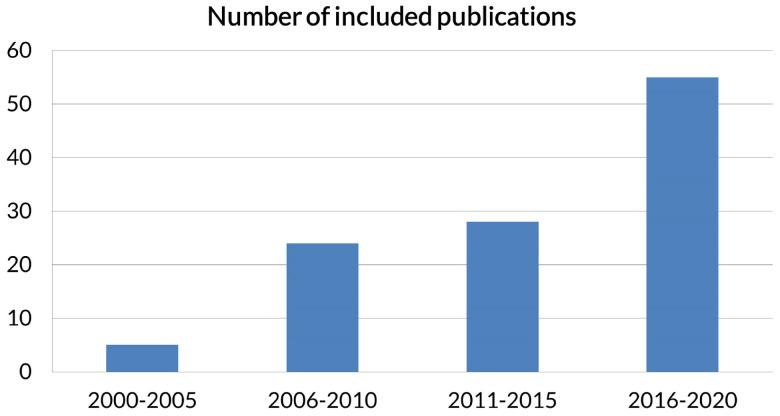
The number of publications per 5-year period.

**Figure 4 sensors-22-04914-f004:**
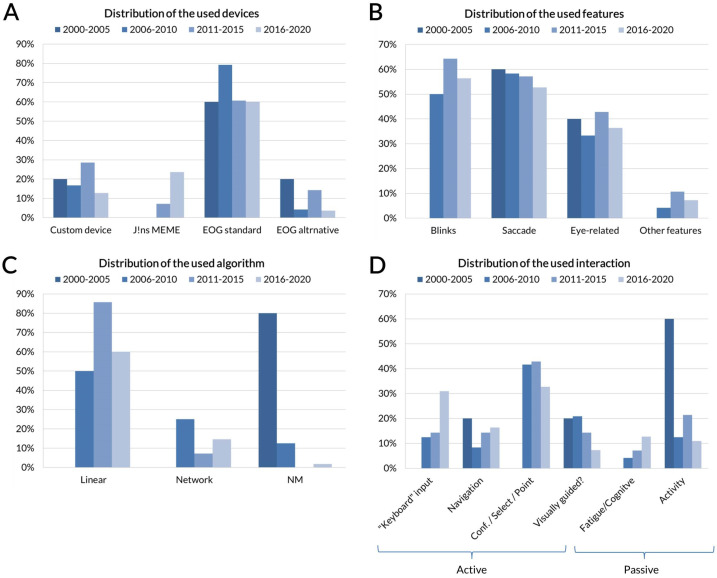
The percentage of references per 5-year period according the (**A**) used device, (**B**) used features, (**C**) used algorithms and (**D**) used interaction.

**Figure 5 sensors-22-04914-f005:**
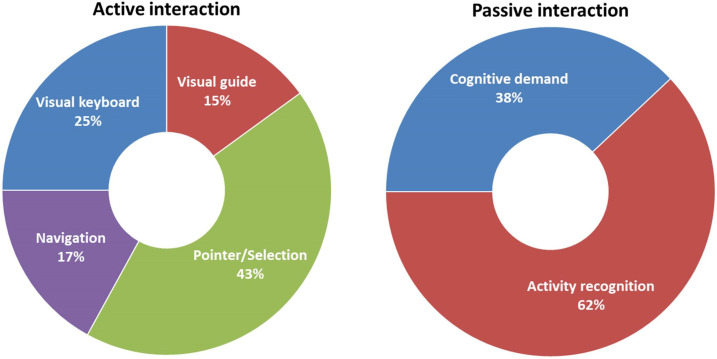
The percentage of each type of active and passive interaction in the included EOG-based BCI publications during the last 20 years.

## Data Availability

Not applicable.

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
