# Peer review of "EOG-Based Human–Computer Interface: 2000–2020 Review"

_sensors, 2022, doi:10.3390/s22134914_

Round 1
Reviewer 1 Report
The paper is an interesting review for general readers.
To make the paper more useful to international researchers in the field, I recommend adding a new section on complexity analysis of the used techniques in this review.
Author Response
We thank R1 for this insightful suggestion. Indeed our paper targets general readers with little knowledge of the EOG technology and data processing technique. Our main contribution regards the collection and the structure of existing EOG research papers. To the best of our knowledge, such a review paper does not exist so far and will be of great use for newcomers in this field. We agree with R1 that such an addition could be worthy but it may be part of another paper due to the broad spectrum of information to be included. Nevertheless, we added in the discussion part high-level direction to highlight the steps for EOG data processing and thus partially showing the complexity of such data analysis.
We also generally improved the manuscript by clarifying the methodology and providing a PRISMA flow diagram, as well as adding a quantitative comparison of different 5-years periods.
Reviewer 2 Report
The considered manuscript reviews publications related to application of electro-oculography in Human-Computer Interaction (Brain-Computer Interfaces), with a particular focus on aeronautics. The authors cover 20 years and over 100 papers, producing a potentially useful overview of the field. However, I see major disadvantages in the methodology, focus, presentation of the results, and the contributions. The organization and formatting of the paper need to be improved also, before I could recommend its acceptance to Sensors. My detailed comments are presented below.
== Methodology ==
First of all, there is a lack of rigor in the considered research work. There are well-known methodologies for undertaking reviews, and even though the authors do not claim that they perform e.g. Structured Literature Review, any scientific research must be reproducible. This, in particular, means that the authors need to clearly describe how they selected the publications they consider in their work. The claims they make should be justified, preferably by quantitative data - e.g., frequencies (like in Fig. 2), keywords analysis, meta-analysis, etc.
== Research question ==
Second (related to the previous one) is the research question, whose formulation is basically missing. What exactly do the authors want to find out about the considered field? Why do they (sometimes) lean towards aeronautics?
I would recommend ending the Introduction with the research question instead of Fig. 2 (I am not sure why Fig. 2 is in the Intro, as it rather belongs to the results). This would help formulating stronger contributions (see below).
== Results & Presentation ==
Although the authors claim that "This paper started with summarizing and organizing the EOG based BCI literature of the last two decades", they do not justify much, why they analyze exactly these properties in the publications ("EOG signal acquisition, EOG device particularity, extracted features, translation algorithms..."). Meanwhile, this appears essential, as the manuscript is organized exactly along these properties used as chapters.
The authors also need to think how to present their intermediate findings in an aggregated form - diagrams, tables, etc. Presenting many papers and briefly retelling their key points are not a scientific review. Tables 1 and 2 are somehow better, but they are still merely descriptive, the authors do not even make comparison between the considered periods.
== Contributions ==
In my opinion, the conclusions are weak, as they are rather general and arguably could be done without relying on formal review of 100+ publications. Moreover, it is not clear how they are supported by the presented results.
For instance, "We conclude that almost all the designed EOG systems are highly promising in terms of performance and do not require heavy signal processing and complex pattern recognition algorithms such as those used with other systems as EEG." - but the authors do not perform quantitative meta-analysis of EOG performance in the considered publications, nor do they do this for EEG systems with which they make the comparison.
Unfortunately, this is even more true for recommendations: "An efficient system should constantly strive to improve speed and accuracy to eliminate the operator’s frustration and to enable a stable performance level. In addition, the new generation models may continually take into consideration the user experience which may reduce the physical and mental stress." - I suppose, any system in any field should strive to improve speed, accuracy, stability and user experience. The authors need to make more specific conclusions and recommendations and clearly show how they follow from the publications that they review.
Additionally, it was claimed in the Abstract that the manuscript "provides a structured design space with a large set of representative papers", but I think I fail to see this in the text.
== Misc and technical comments ==
Why do references start with [150]?
The References section is included twice, but the items there seem to be the same.
If the authors choose to use present tense in the Abstract ("First, we highlight", "Second, we provide"), it should be used consistently:
"We concluded" -> "We conclude"
"we provided suggestions" -> "we provide suggestions"
Mistype in Table 1: Reference
"Author" should be actually "First author", to avoid discounting the work of the co-authors.
The font size in Fig. 2 should be increased.
Author Response
We thank R2 for a thorough review of the manuscript, it was greatly helpful to improve the quality of the paper.
Regarding the methodology, we added more details on the research request and the PRISMA flow diagram for the review process.
Regarding the research question, results, and presentation, we agree with the reviewer and moved Fig 2 and an associated paragraph further in the manuscript in the results section. We also added two figures for quantitative comparison between 4 successive 5-year periods.
Regarding the references, many thanks for noticing this issue. It appears to be a text processing issue with the pdf generation by overleaf, it is the generated file on my computer the reference started from [1]. It should be corrected now.
Also many thanks for the misc and technical comments, we corrected the tenses, and table 1 & 2 headers as well as updated Fig2 so that the fonts are bigger.
Reviewer 3 Report
The article presented by the authors for review is an attempt to present the current state of knowledge regarding the use of EOG signal as a basis for human-computer interfaces. The authors have undertaken a difficult task. The EOG signal is an electrophysiological signal that is recorded on the surface of the skin. This leads to similar problems as in the case of other electrophysiological signals (ECG, EEG, EGG, EMG, ENG or others) - non-stationarity of interference, problem with electrode placement and their properties. The authors are aware that the information obtained from the EOG signal is not ideal, if only because of the low vertical and horizontal angular resolution, but they present the possibilities of using this signal in the human-computer interface. This is evidenced by a total of over 200 references cited by the authors.
Yet some of the authors' statements may be incomprehensible. I will try to list a few.
1. line 64: "... we identified two main interactions: active and passive interaction"
What specifically do the authors mean? If to describe it briefly.
2. Line 206: "After that, an averaging filter is used to eliminate the existing noise components" here I think we rather mean moving averaging filter, because what is marked can be understood as an averaging filter for pseudo-periodic signals. This needs to be clarified.
3. line 236: "The minimum redundancy and maximum algorithm analyze the feature that associates the strongest feature with the classification feature and combines it with mutually different selection features." What did the authors mean by this?
4. line 319-326: This paragraph does not fit here in my opinion. It contains interesting content, but in my opinion it should be elsewhere in the work.
Technical faults
1. reference list, item 157 - cannot be downloaded,
2. line 182: flirted ???,
3. bizarre way of numbering the references,
4. line 293, 294, (eg, ...) should be, no punctuation marks.
Author Response
We appreciate the thorough review of our work and thank the reviewer for the suggestions.
- We added two phrases and a reference regarding active/passive interactions
- We corrected that we meant the moving average filter (thank you for pointing that out)
- We corrected the technical bugs and removed the reference that cannot be downloaded (It appears that the server is no longer responsive; we’ve removed the reference because it was cited among the other three papers and is not essential. Thank you for the remark.)
- Regarding the references, it was an overleaf bug, thank you for noticing, it should be correct now.
- We added punctuation throughout the manuscript.
Round 2
Reviewer 2 Report
I have read the authors' reply and the updated version of the manuscript. I can see significant improvements in the methdological aspect, as Fig. 2 now clarifies it pretty well. I also commend the authors for enhancing the presentation of the results. Unfortunatelly, there seems to be no changes in the Conclusion section, and I still encourage the authors to think if they can extend and detail it.
Still, I find this remaining issue to be minor, and can already recommend the manuscript for acceptance.
Author Response
We thank the reviewer and appreciate their effort in the manuscript review. We updated the conclusion accordingly and added a new paragraph to underline the manuscript revisions and the results in the conclusion section. The changed part is highlighted in magenta.